# *What If...?* Pandemic policy-decision-support to guide a cost-benefit-optimised, country-specific response

**Giorgio Mannarini**[1], **Francesco Posa**[1], **Thierry Bossy**[1], **Lucas Massemin**[1], **Javier Fernandez-Castanon**[2], **Tatjana Chavdarova**[1,3], **Pablo Cañas**[1], **Prakhar Gupta**[1], **Martin Jaggi**[1]*, **Mary-Anne Hartley**[1]*

**1** Intelligent Global Health, Machine Learning and Optimization Laboratory, EPFL, Lausanne, Switzerland, **2** Independent Researcher, London, United Kingdom, **3** University of California Berkeley, Berkeley, CA, United States of America

☯ These authors contributed equally to this work.
* mary-anne.hartley@epfl.ch (MAH); martin.jaggi@epfl.ch (MJ)

**Data Availability Statement:** All the data sources and their availability is stated in S1 Table in the manuscript. SwissRE is a tertiary source of these

## Abstract

### Background

After 18 months of responding to the COVID-19 pandemic, there is still no agreement on the *optimal* combination of mitigation strategies. The efficacy and collateral damage of pandemic policies are dependent on constantly evolving viral epidemiology as well as the volatile distribution of socioeconomic and cultural factors. This study proposes a data-driven approach to quantify the efficacy of the type, duration, and stringency of COVID-19 mitigation policies in terms of transmission control and economic loss, personalised to individual countries.

### Methods

We present *What If...?*, a deep learning pandemic-policy-decision-support algorithm simulating pandemic scenarios to guide and evaluate policy impact in real time. It leverages a uniquely diverse live global data-stream of socioeconomic, demographic, climatic, and epidemic trends on over a year of data (04/2020–06/2021) from 116 countries. The economic damage of the policies is also evaluated on the 29 higher income countries for which data is available. The efficacy and economic damage estimates are derived from two neural networks that infer respectively the daily R-value ($R_E$) and unemployment rate (*UER*). Reinforcement learning then pits these models against each other to find the *optimal* policies minimising both $R_E$ and *UER*.

### Findings

The models made high accuracy predictions of $R_E$ and *UER* (average mean squared errors of 0.043 [CI95: 0.042–0.044] and 4.473% [CI95: 2.619–6.326] respectively), which allow the computation of country-specific policy efficacy in terms of cost and benefit. In the 29 countries where economic information was available, the reinforcement learning agent

datasets in the form of a convenient data stream, but access to this data stream is not essential for reproducibility. The tertiary access to these datasets via SwissRE allowed us free access to Moody's. We had no additional privileges. Moody's and WWO would be paywalled for regular users as indicated in Table S1. We were regular users of WWO.

**Funding:** The authors received no specific funding for this work.

**Competing interests:** The authors have declared that no competing interests exist.

suggested a policy mix that is predicted to outperform those implemented in reality by over 10-fold for $R_E$ reduction (0.250 versus 0.025) and at 28-fold less cost in terms of *UER* (1.595% versus 0.057%).

## Conclusion

These results show that deep learning has the potential to guide evidence-based understanding and implementation of public health policies.

## Introduction

The unprecedented speed and scale of the COVID-19 pandemic necessitated rapid implementation of untested public health measures to mitigate the consequences of viral spread [1]. These policies created massive collateral economic damage from which it is predicted to take decades to recover [2], especially for low-resource settings and marginalised populations [3]. When selecting mitigation strategies, the *optimal* trade-off between lives vs livelihoods (i.e balancing the life-saving benefits of mitigation strategies vs their livelihood-damaging economic costs) is not obvious [4], and perceptions differ according to a volatile mix of socioeconomic, demographic and cultural features that vary between countries and evolve over time. There is a growing appreciation that decisions based on such vastly complex and dynamic data require the advanced pattern detection of deep learning networks [5].

Indeed, the torrent of information generated during COVID has been described in epidemic terms, where the growth rate of scientific publications rivalled the virus itself. Several large scale data trackers have attempted to validate and synthesise the data into massive open-access live-stream global repositories (e.g. for policies (OxCGRT [6]) and proxies of their efficacy such as viral transmission and deaths (Johns Hopkins [7], ourworldindata [8, 9]) or mobility (Google [10])). Palantir's Foundry, designed to curate and integrate vast live data streams, is used in this study to integrate and transform such data sources into a single unified data asset for analysis, modelling, and decision-making. [11]. Nevertheless, mining this data remains complex, and only few studies have attempted to provide models for policy decision support. Furthermore, they are mostly limited in their scope, often only focus on single countries [12–14], single policies [15, 16], single impact measures [17] or a finite time span [18]. Many do not make use of machine learning for updatable insights compatible with real time data streams and most only provide retrospective analyses, associating policies to proxies of the assumed impact.

Inferring causality from such models (for example, attributing the implementation of policy *x* to an impact in a proxy measure of efficacy) is not straightforward. The longitudinal nature of the problem embeds time-dependent confounders that may contaminate the assignment of causality, where the duration or sequence of implemented policies erode or synergise their univariate impact [19]: an issue that would bias the computation of counterfactual analysis proposed by this study.

This work attempts to provide evidence-based policy-decision-support (PDS), not only to tailor responses to the specific contexts of individual countries, but also to optimise them according to a target cost-benefit trade-off.

## Materials and methods

### Study design

This study leverages a diverse global data stream to build two predictive models that infer the impact of the two measures on which we aim to create a trade-off, i.e. viral transmission (effective reproduction number, $R_E$) versus economic cost (unemployment rate, *UER*). A reinforcement learning agent (RLA) then pits the two models against each other to find the policy combination that best minimises both metrics.

The RLA recommends *optimal* policies for a simulated month, with the allowance of generating weekly policy changes in 1-step stringency increments or decrements. The outputs can be adapted to customisable counterfactual *What If. . .?* scenarios, where one can specify the RLA's optimal trade-off between $R_E$ and *UER*. Finally, the predictive models are also explored with a derivation of SHAP values [20] for deep learning models, which are used to infer the impact of each policy for each country on the predicted measures.

### Data sources

The start date of data used aligns with the WHO pandemic declaration and spans over a year from 01/04/2020–31/05/2021. Predictions of both models are made on a temporal scale of a day and at the geographic resolution of a country. While this study relies on a uniquely diverse live global data stream that was curated by the Swiss Re Risk Resilience Center [21], it is sourced from benchmark sources as summarised in S1 Table.

S2 Table reports the features used in each model. Only features collected before (and including) the date of the predicted value are considered in the prediction (thus the model only uses past data to predict into the future). The data considered includes:

- Standardised COVID Policies features: The type, duration and stringency of global COVID mitigation policies implemented in each country as curated in the Oxford COVID Government Response Tracker (OxCGRT) [6], which comprises 12 policy types (*cancel public events, close public transport, testing level, contact tracing, vaccination policy, international travel controls, public information campaigns, gathering restrictions, internal movement restrictions, school closing, stay at home requirements, and workplace closing*) with date-stamped implementation histories categorised into stringency levels normalised across 186 countries.

- Country-specific characteristics features: A set of socioeconomic (e.g GDP), demographic (e.g. total population stratified by age) and climatic (e.g temperature) features. The two models ($R_E$ and *UER*) made use of slightly different combinations of these features, where for example unemployment rate was included in the $R_E$ model as a predictor but excluded from the *UER* network as it represented the predicted label. Similarly, no epidemic features were used in the $R_E$ model, but the measured $R_E$ was used as a feature in the *UER* model. These features are available for 168 of the 186 countries included in the OxCGRT dataset.

- $R_E$ label: The daily $R_E$ is the ground truth of the first model and was obtained following the formula described by Abbott et al. [22], with a moving average of 12 weeks. A reporting artefact occurred in the presence of extremely low case numbers (generally at the beginning of the pandemic), resulting in an artificially inflated $R_E$ value. To limit this effect, we discarded the estimations of $R_E$ that were greater than 4, which was epidemiologically implausible according to the transmission dynamics of the initial wild type and alpha variants reported to be in circulation at the start of the pandemic [23].

- *UER* label: We linearly interpolate quarterly reported/forecasted values of *UER* to obtain daily estimates. Forecasted values are predicted by unspecified means by the cited source in S2 Table. We then predict the *UER* deviation compared to the baseline measurement at the beginning of the pandemic (31/12/2019).

## Country selection

The list of countries for each model ($R_E$ and *UER*) is available in S3 and S4 Tables.

**Geographic scope of the $R_E$ model.** Of the 186 countries represented in the OxCGRT dataset, 168/186 (90%) had the required socioeconomic features. Of these, 116/168 (69%) met the inclusion criteria of the $R_E$ model described hereafter (excluding 52 countries). Firstly, countries with fewer than 2000 COVID-19 cases were removed (n = 6) as such data sparsity is not informative to the model and is sometimes indicative of poor case reporting such as Tanzania, which stopped reporting in May 2020 after just 500 cases. The remaining 46/52 countries were excluded for not having available data for the features required by the predictive models. The retained countries represent roughly 90% of the global population across a wide variety of sociodemographic settings from each continent. It covers a GDP range of 1.4 to 17900 billion USD (measured on 31/12/2020) with an average urban population of 60% (compared with 64% globally according to the World Bank data sources).

**Geographic scope of the *UER* model.** Of the above selected countries, only 29/116 (25%) provided suitable labels that could be validated for the *UER* model. Indeed, unemployment rate is more challenging to collect and data quality standards (absence of null values, values up to 31/03/2021 collected and not forecasted), were only met in few countries. To ensure that we limited assumptions to the most represented subset, we focused on high income settings within close geographic proximity as a proof-of-concept, including 27 states from the European Union as well as neighboring Switzerland, Norway and the UK. When applying the same filters as above, only Poland was excluded due to missing key features. Thus, the economic component of this work is limited to higher income countries. We hope the results can serve advocate for the potential value of regular economic reporting in lower resource settings. This work is open source so that new data can be added as it becomes available.

## Model architecture

The code for all models is available at this link.

The features used to predict $R_E$ and *UER* are a mixture of time-dependent (e.g. weather) and time-invariant (e.g. GDP) features. To deal with this mixed-type data, we created a hybrid deep learning model combining a recurrent neural network in the form of a Long short-term memory (LSTM) [24] and a multilayer perceptron (MLP). This latter fully connected part takes the constant features as input while the LSTM was employed to analyse sequential trends (Fig 1). More specifically, the two-headed network consists of:

- *First head*. One LSTM layer, reserved for time-dependent features, with a hidden size of 20.

- *Second head*. One linear (fully connected) layer of 50 neurons for the time-invariant features, followed by a ReLU activation function.

- *Body*. Two linear layers with two ReLU activation functions. The first layer has 15 neurons, the second (output) has one. The ReLU on the output is applied given the nature of the problem (negative $R_E$ or *UER* are impossible).

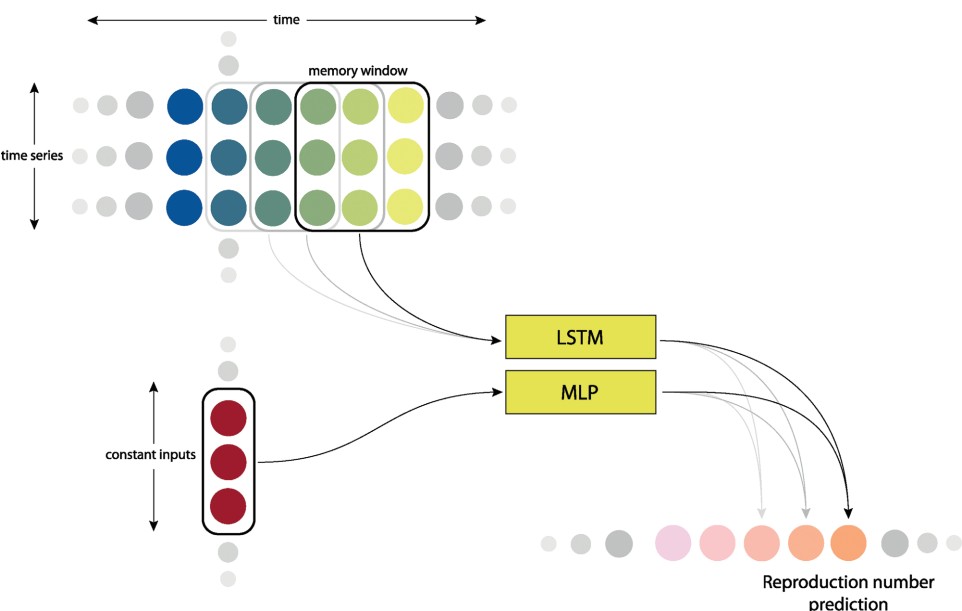

**Fig 1. Neural network architecture.** Architecture of the hybrid neural network model for predicting $R_E$. Constant inputs denote the time-invariant input features.

To reduce over-fitting, a dropout layer is placed after each layer with a dropout probability of 0.2. In addition, we employed early stopping conditional to loss stagnation for five epochs. Here, the model restores the weights captured from the epoch corresponding to the best validation loss value.

The memory window of the LSTM layer can be parameterised in the number of days. A memory window of $n$ days means that to predict the output at a day $d$, the time series corresponding to the sequential feature values on days $d - (n + 1)$, $d - n$, . . ., $d - 1$ and $d$ are used as inputs. The $R_E$ model uses a memory window of seven days, where the forecasting horizon is zero, as we use data from seven days before day $d$ to predict only the $R_E$ of day $d$. In contrast, the *UER* model uses a widened memory window of 28 days. As described above, the *UER* is a quarterly measure which was interpolated to daily point estimates. To ensure the actual measured *UER* have a higher importance than the interpolated ones, a weight between zero and one was assigned to each interpolated value according to their proximity to the true reported value (where the weight is linearly eroded to zero according to the distance from the nearest measurement date). This gradual reduction of "importance weighting" for interpolated values further away from the date of the actual measurement, reduces the likelihood of error introduced by the linear interpolation while providing the models with more training data.

## Model validation

A leave-one-out cross-validation strategy was used to validate the performance of our models, where each fold corresponds to a unique country, i.e., for each validation step, a different country is isolated, the model is then trained on the data from all other countries and then validated on the isolated country. These cross-validation subsets, allow the model to predict the evolution of a feature in one country by using data from other "similar" countries. It also better accounts for temporal confounders where the duration or sequence of implemented policies erode or synergise their univariate impact. For instance, if lockdown was implemented several

days after a gathering ban, the effect of lockdown might be diluted. Thus, our model would learn from similar contexts where policies were implemented in different orders, allowing more reliable estimates of policy impact as described below.

## Policy impact estimates

Of the twelve policies in the OxCGRT dataset, nine are explored for their impact on $R_E$. Two policies were not considered (*Testing level* and *Contact tracing*) due to their power to modulate case reporting i.e. fewer tests/contact tracing results in an artificially lowered $R_E$ due to lower case discovery which makes impact estimates on these policies pointless. *Vaccination policy* is excluded as our aim was to estimate the impact of *non-pharmaceutical* interventions.

To estimate the policy impact in terms of $R_E$, we relied on expected gradients [25], which approximate SHAP values [20] for deep learning models. The purpose of this algorithm is to map the final prediction on the input features in terms of impact, indicating the quantitative contribution of features to the output value.

For a single country, the $R_E$ was computed for *D* different days. With a standard MLP architecture, this means that expected gradients would return a $D \times F$ matrix, where *F* is the number of features. To have a single, global importance for each feature, we would then average on the rows of this matrix. However, in our case, we were interested in computing the importance of each policy, which is variable over time and thus is fed into the LSTM layer of our hybrid model. This means that, considering a memory window of *n* days, the output shape of expected gradients is $D \times n \times F$ (where *D* is equal to seven (consecutive days) in this case, being the memory window of the LSTM layer), as each feature value contributes to *n* predictions. Fig 2 better explains this concept: for a single feature (policy), the output is a $D \times n$ matrix. The dots of the same color represent the fact that a single value is taken into account in *n* different predictions. Given that, we averaged over the diagonals of this matrix, computing the importance of each unique value of the considered policy. We then computed a global average to obtain a single feature importance. Finally, the values are normalised to a min-max [0; 1] interval, where 0 represents the lowest possible importance.

## Reinforcement Learning Agent (RLA)

The RLA seeks the best combination of non-pharmaceutical policies to minimise both $R_E$ and *UER* given a country's epidemic and socioeconomic situation. To achieve this, the agent is trained on the predictions of the two networks presented above. In particular, the agent models the impact of various permutations of policies at various stringency levels and receives a reward proportional (in percentage) to the reduction of either $R_E$ or *UER*, as shown in Algorithm 1.

Some modifications to the features used in both networks of this task were necessary; since, for example, it is impossible to predict future weather with high accuracy, these features were thus excluded from the training of the two $R_E$ and *UER* networks. We show in Section *$R_E$ and UER models used in reinforcement learning* that this change does not affect performances.

All vaccination features were also eliminated for the same reason.

**RLA constraints.** *Action space*. With nine policies at four stringency levels, the number of permutations in the RLA action space for a single decision is too large (approximately $4^9$) for standard RL algorithms such as Policy Gradient or Deep Q-Learning (DQN).

We thus re-examined the problem from a continuous perspective viewing each policy as a continuous number, with the output of the RLA being between −1 and 1. We then add this vector to the policy vector of the week before to obtain the new set of policies.

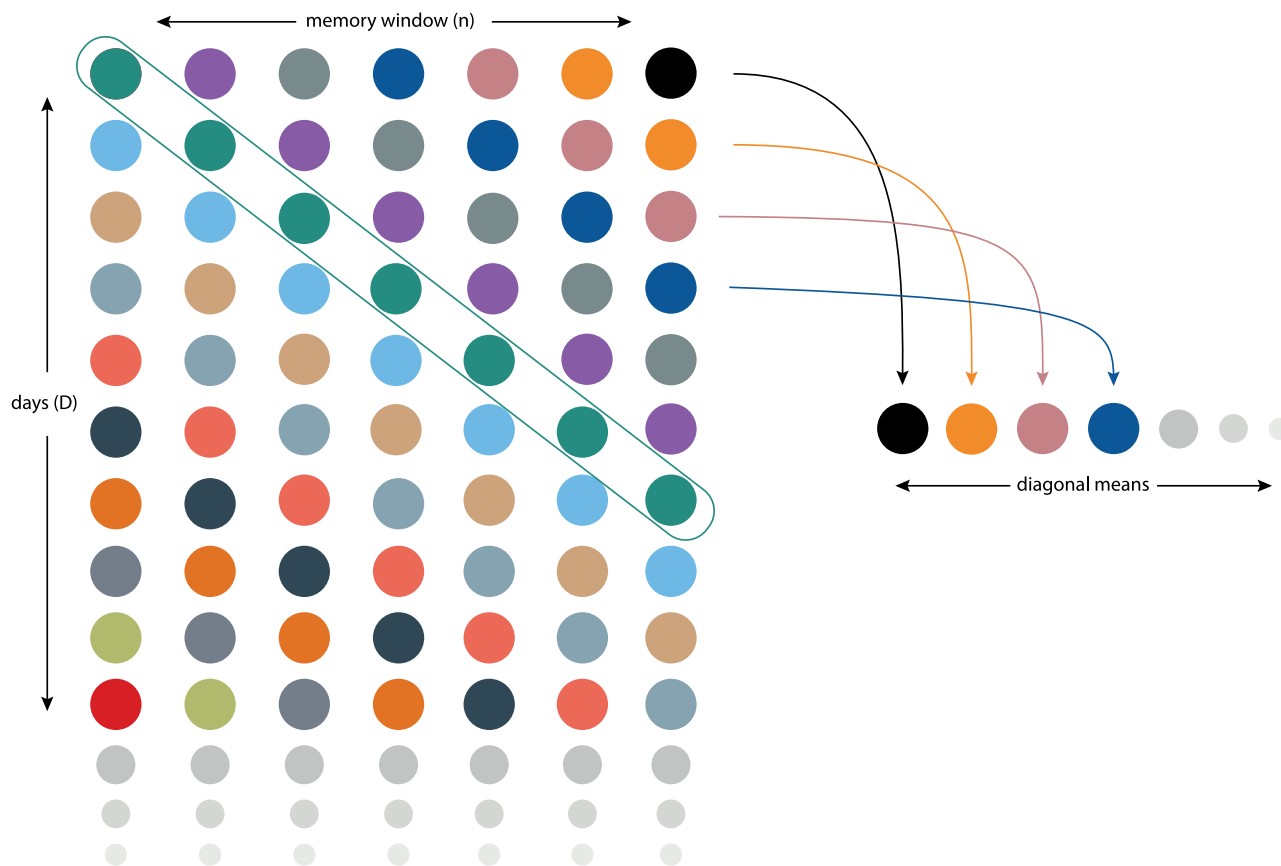

**Fig 2. Diagonal mean—Feature importance.** Mean of diagonal means to compute a single importance value for a single feature in the context of an LSTM model.

As the final policy mix is continuous, the network may set a policy to a negative value or a value that is out of range according to our data. Because it is not realistic to set policies to a negative state (e.g. there is no state of "anti-lockdown"), the policies are bounded between 0 and the highest value of the policy according to OxCGRT dataset. The policies are also rounded to integers for better interpretability and standardisation.

*Policy change frequency*. In order to reduce the model's variability, we permit the agent to change policies every seven days over a 28-day period. Thus, the model makes four predictions each 28 days, proposing the best set of policies to apply for the following weeks.

*Recommended policy variability*. To ensure a *sensible* variability to policy recommendations the agent is restricted to making 1-notch stringency increments or decrements in policies each week. This means that if in week *w* policy *p* had a value of *x*, in week *w+1* this value will be in a range between $[x - 1, x + 1]$. This avoids the unrealistic scenario of having a strict lockdown in one week and then no restrictions at all in the next.

*Trade-off threshold*. The RLA explores the policy space, receiving rewards proportional to how much a specific policy mix minimises the two indicators ($R_E$ and *UER*) together, in percentage with respect to the starting value. While the current work gives these equal importance, it is possibile to give more weight to one of the two indicators with respect to the other.

**RLA architecture.** The network's input consists of the set of policies adopted during the previous four weeks with respect to the prediction. This means that the network computes the

optimal policy set for the first week in the future given the policies adopted during the last four weeks and the associated $R_E$ and *UER*, which are used to compute the rewards of the agent. For the second week, the input will consist of the actual policies adopted in the last three weeks plus the predicted policies of week one, and so on. Thus, the state space size is a $4 \times 9$ matrix, which is then flattened before being fed to the network. The RLA is based on a Deep Deterministic Policy Gradient (DDPG) algorithm. Algorithm 1 shows the model training loop, given the DDPG agent and the considerations made before. Our results are obtained training the model for 5000 episodes for each target country. In particular, we adopt the same leave-one-out-validation methodology: the $R_E$ and *UER* models are trained on all the countries except for the target one, and their predictions are then used in the training of the agent.

**Algorithm 1**: RL agent training loop

```
Result: Best policy mix for each week (sₜ)
Select a country;
Train the reproduction rate (Rₑ) and unemployment rate (UER) networks
on all the other countries;
Initialize the DDPG agent with state space and action space sizes σ
and α;
Initialize the Rₑ weight ωᵣ ∈ [0, 1] and the UER weight ωᵤ = 1 – ωᵣ;
for episode = 1, M do
  Set sₜ to the policy levels of the last 4 weeks for the selected
country;
  Predict the initial Rₑ and Uᵣ given sₜ;
  for step = 1, number of future weeks do
    Select action aₜ = agent(sₜ);
    To create the new state sₜ₊₁ delete the first α elements from sₜ
andreplicate the last α elements at the end of sₜ. Then, sum aₜ to the
last Predict Rₑ₊₁ and UER₊₁ from sₜ₊₁;
    Set reward rₜ₁ to (Rₑ−Rₑ₊₁)/Rₑ · 100;
    Set reward rₜ₂ to (UER−UER₊₁)/UER · 100;
    Set rₜ = ωᵣ · rₜ₁ + ωᵤ · rₜ₂;
    Store transition (sₜ, aₜ, rₜ, sₜ₊₁) in the agent memory;
    Train the agent;
    Set sₜ = sₜ₊₁;
  end
end
```

## Results

### $R_E$ model

Using all features in S2 Table, we obtain an average MSE (aMSE) of 0.043 with a 95% confidence interval of [0.042, 0.044] for the prediction of $R_E$ across all 116 countries considered. The MSE of each country is reported in S3 Table. Fig 3 shows the predicted $R_E$ for three randomly selected example countries on three different continents compared to the ground truth of the reported $R_E$ estimation (Switzerland (CHE), Morocco (MAR) and Peru (PER)). As observed, the predicted $R_E$ follows quite closely to the ground truth value in all three countries despite the vastly different combination of mitigation policies implemented as well as the underlying differences in geography, culture and socioeconomic context. These results are *forecasted*, where the model uses past data to make predictions over a 7 day period.

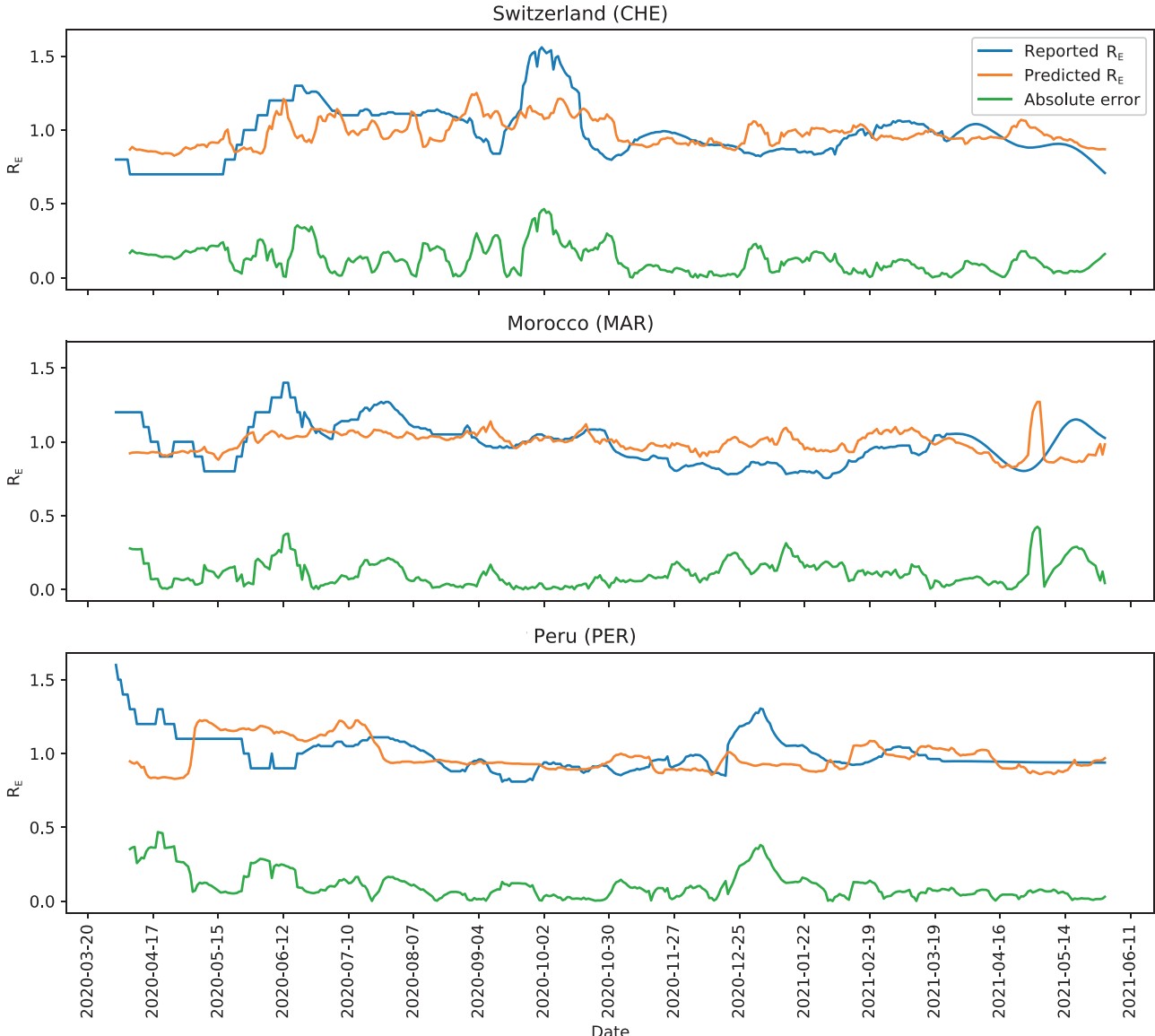

**Fig 3. Reproduction rate predictions for three sample countries.** $R_E$ prediction (orange) for Switzerland (CHE), Morocco (MAR) and Peru (PER). The ground $R_E$ is shown in blue, and the absolute error (absolute difference between the prediction and ground truth) in green.

## Policy impact

The full list of policy impacts in terms of SHAP values for the nine policies across all 116 countries is presented in a heat map in Fig 4. We observe the anticipated result that each policy has a different impact ranking in relation to the context in which it was applied.

To illustrate our claim, we show the policy impact rankings of the same three example countries as above Fig 5. A stark example of the differences of policy effectiveness between countries is seen in "Lock Down" (*Stay at Home Requirements*), which proved highly effective in Switzerland and Morocco, but was only minimally associated with decreased transmission in Peru. The effectiveness of lockdown-type measures in Switzerland and Morocco (and the failure of lockdown in Peru) are supported by independent studies [26–28].

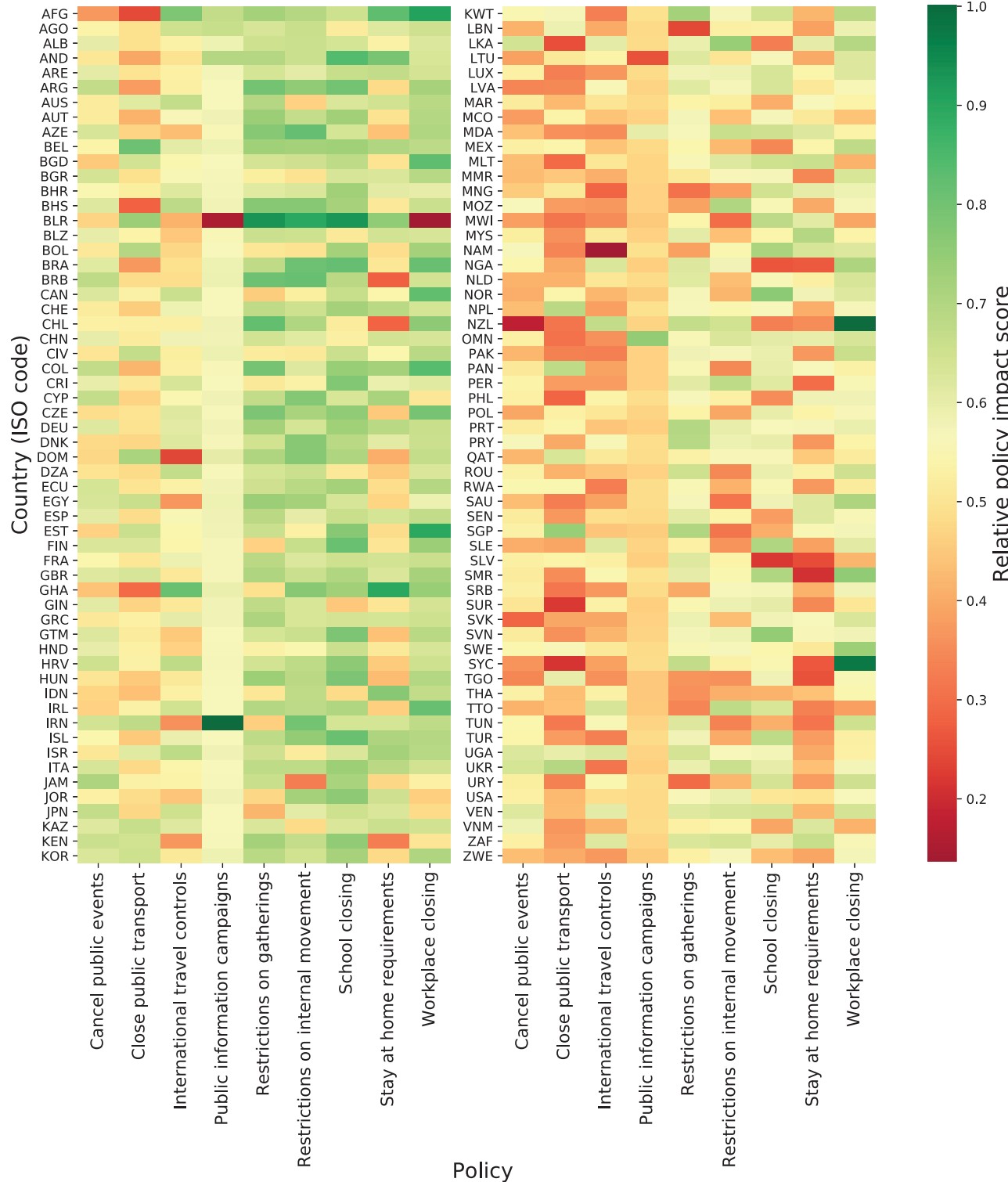

**Fig 4. Policy impact score—116 countries.** Relative policy impact score for 116 countries and nine policies. It represents how much a policy affects the $R_E$ on each country on average. The more red the color, the lower the impact.

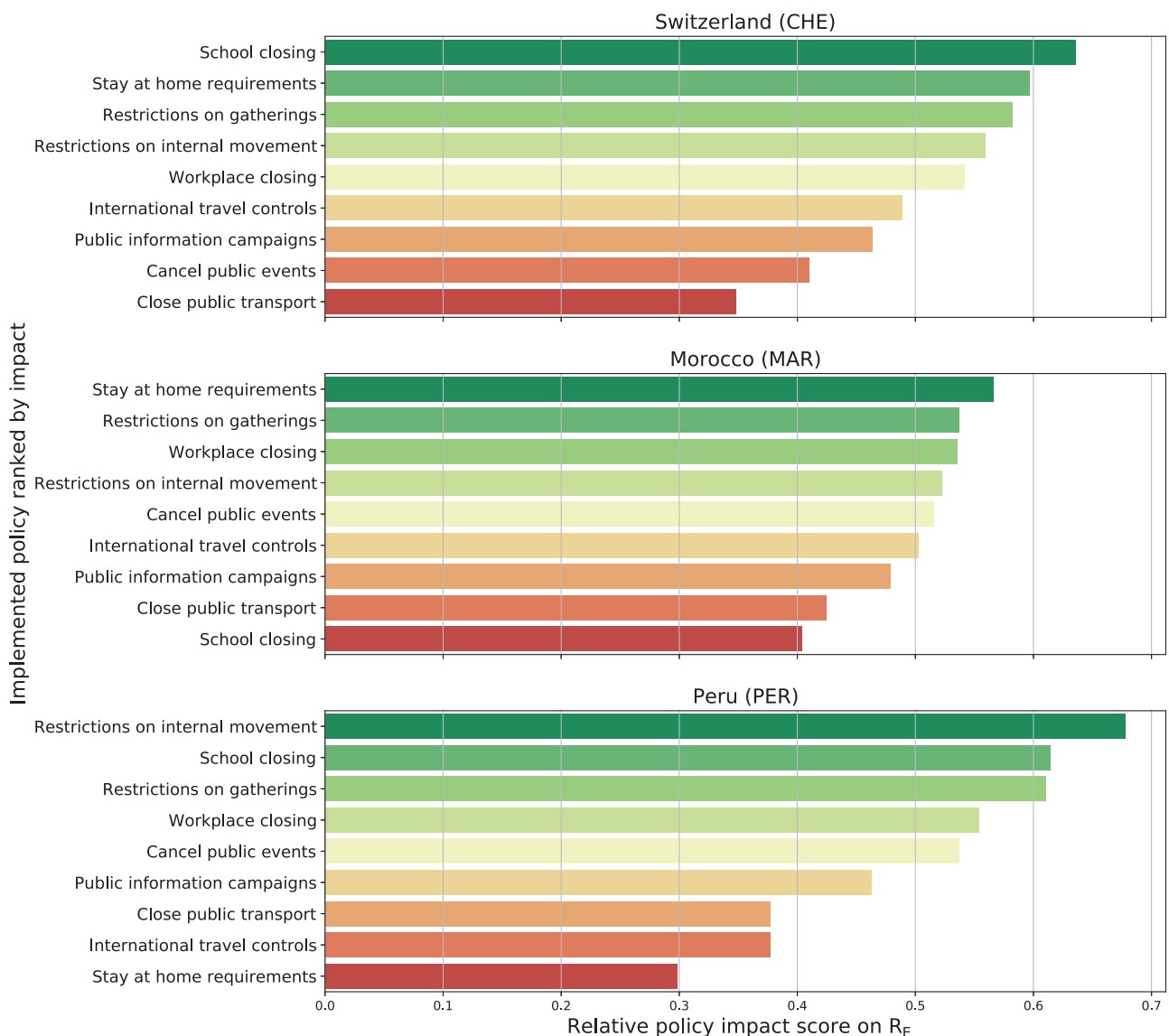

**Fig 5. Policy impact for three sample countries.** Relative policy impact for (top to bottom) Switzerland, Morocco and Peru. An high impact score (green bars) is associated to a reduction of the $R_E$.

## Unemployment rate model

Using the features listed in S2 Table, we obtain an average MSE (aMSE) of 4.473 percentage points with a 95% confidence interval of [2.619, 6.326] over the 29 countries considered in predicting *UER*. Fig 6 shows the predicted *UER* for three randomly selected example countries compared to the ground truth of the reported (and interpolated *UER* estimation (Switzerland (CHE), France (FRA) and Denmark (DNK)). As observed, the predicted *UER* follows quite closely the ground truth in all three countries despite a vastly different combination of mitigation policies implemented as well as the underlying differences in geography, culture and socioeconomic context. There are, however, errors in the prediction (notably CHE), which are likely attributable to gaps in the predictive capacity of the model, but may also be an indication

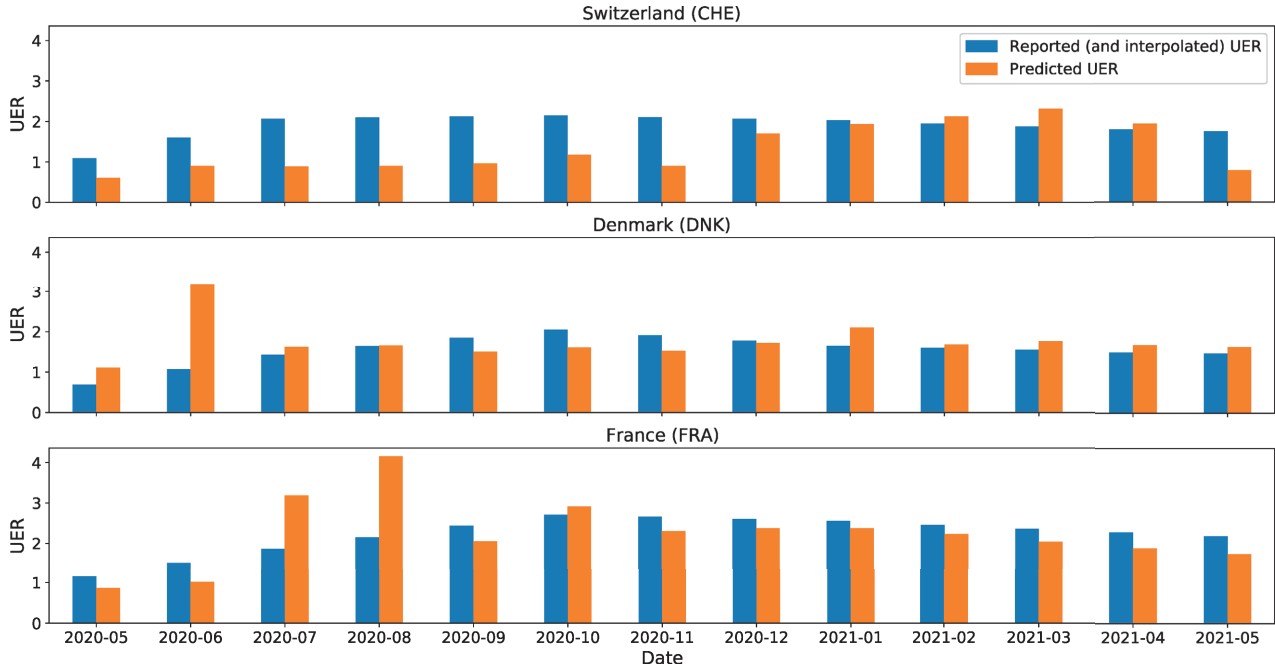

**Fig 6. Unemployment rate predictions for three sample countries.** Unemployment rate predictions for (top to bottom) Switzerland, Denmark and France. The ground unemployment rate is shown in blue, the predicted unemployment rate in orange.

of whether the country performed as "expected" compared to other "similar" countries. This is a possible conclusion due to the leave-one-out-validation strategy, which computes error as compared to other countries. For example, if a country had predictions of *UER* below the real value, this could indicate that the country suffered more than expected, and vice-versa.

The MSE of each country is reported in S4 Table, where we highlight a significant outlier, Spain, with an MSE of 25.1380 percentage points, perhaps indicating that more diverse and accurate data is needed, in addition to the considerations mentioned above.

## Predicting the best policy combination

**$R_E$ and *UER* models used in reinforcement learning.** The reinforcement learning model uses the two networks predicting $R_E$ and *UER* trained on feature set listed in S2 Table. Given the leave-one-out-validation methodology, there is no leakage between the two networks training set and the agent, which is rewarded according to the predictions made on the testing country. The minor feature set changes had little impact on individual model performance, where the global MSE was 0.0416 CI95 [0.032, 0.051] for $R_E$ and 4.4495 CI95 [3.645, 5.254] for *UER* (compared respectively to 0.043 CI95 [0.042, 0.044] and 4.473 CI95 [2.619, 6.326] for predictions using the original feature sets). A two-sample t-test confirms that the differences in the means are not significant (the p-values are respectively 0.771 and 0.988). S5 and S6 Tables show the MSE when predicting the $R_E$ and *UER* for the 29 considered countries with the new feature sets.

**RLA versus human policymakers.** To evaluate our model in terms of the efficacy of policies selected by (human) policymakers in reality versus those recommended by our RLA, we compute the $R_E$ and *UER* trends predicted for the counterfactual scenario of implementing the policies recommended by our RLA for each of the four weeks. We select random time points on which to test these trends.

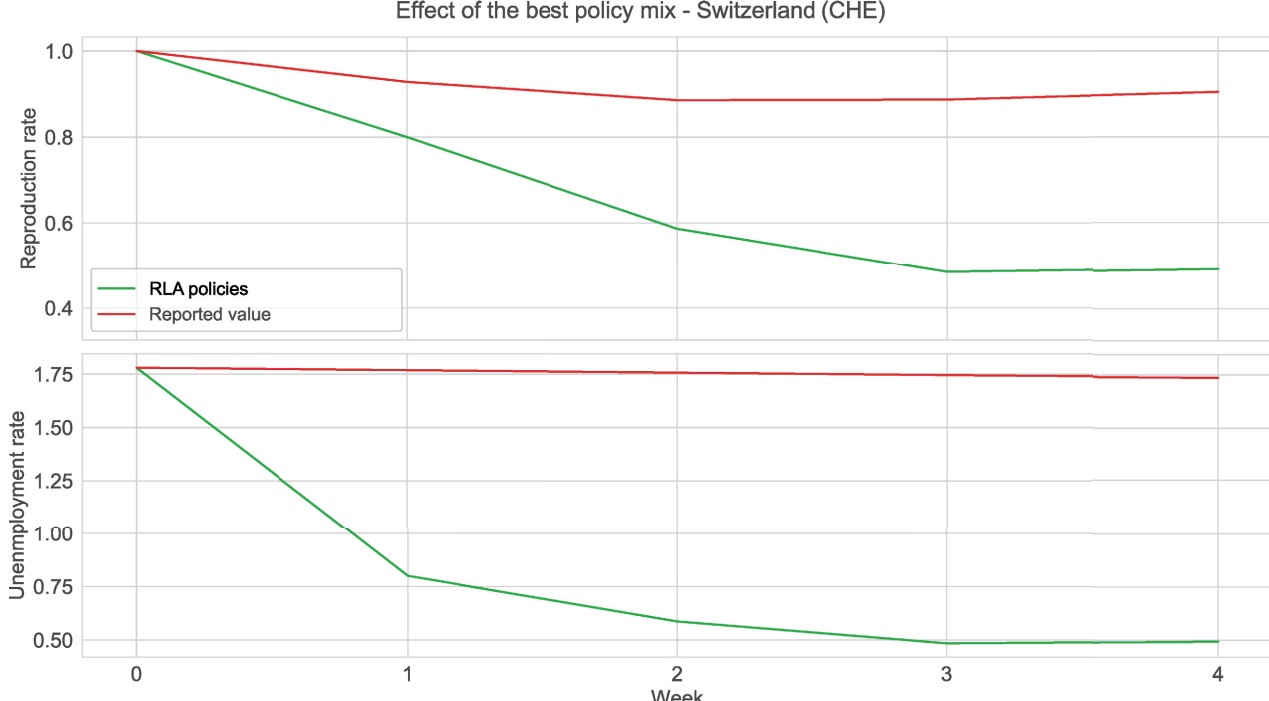

**Fig 7. Effect of the best policy mix on Switzerland.** Effect of the best policy combination on the $R_E$ and *UER* (12/04/2021–09/05/2021). The green line refers to the best policies predicted by the RLA. The red line is the true, measured, value.

S7–S9 Tables list differences in policy impact between human policymakers versus the RLA for all 29 countries for three randomly selected time periods. S10 Table shows the mean of these differences.

In our simulations, the RLA has the same reward threshold for $R_E$ and *UER*. In a randomly selected one month time span (12/04/2021–09/05/2021), the average difference between the reported $R_E$ and the predicted $R_E$ when adopting the RLA-suggested policies is 0.211. As for the *UER* this difference is of 1.393 percentage points. Thus, the RLA policies are predicted to reduce $R_E$ and *UER*. The simulation on the other two randomly selected time spans follow the same trend, with better results for the two metrics than what was implemented in reality.

As a proof of concept, we show the weekly effect of the best policy mix for Switzerland (Fig 7) and the related policy levels that the network suggested (Fig 8) in the time period of 12/04/2021–09/05/2021. The results are promising as we were able to significantly predict a reduction in both the pandemic $R_E$ and the unemployment rate with respect to the starting situation without necessarily enforcing the highest level for each restriction. The recommended policies in Fig 8 are interpretable and implementable. We can see that the RLA recommended highly stringent workplace closures along with intensive international travel controls, and a moderate stay-at-home requirement in addition to minimal restrictions on public transport, gatherings and events. This is a contrast to the implemented policies in Switzerland at the time, which included a higher level of restrictions on gatherings, but a lower level of workplace closing.

## Discussion

This work explores how the public health policies implemented during the COVID-19 pandemic were related to country-specific patterns in the reported metrics they were presumed to

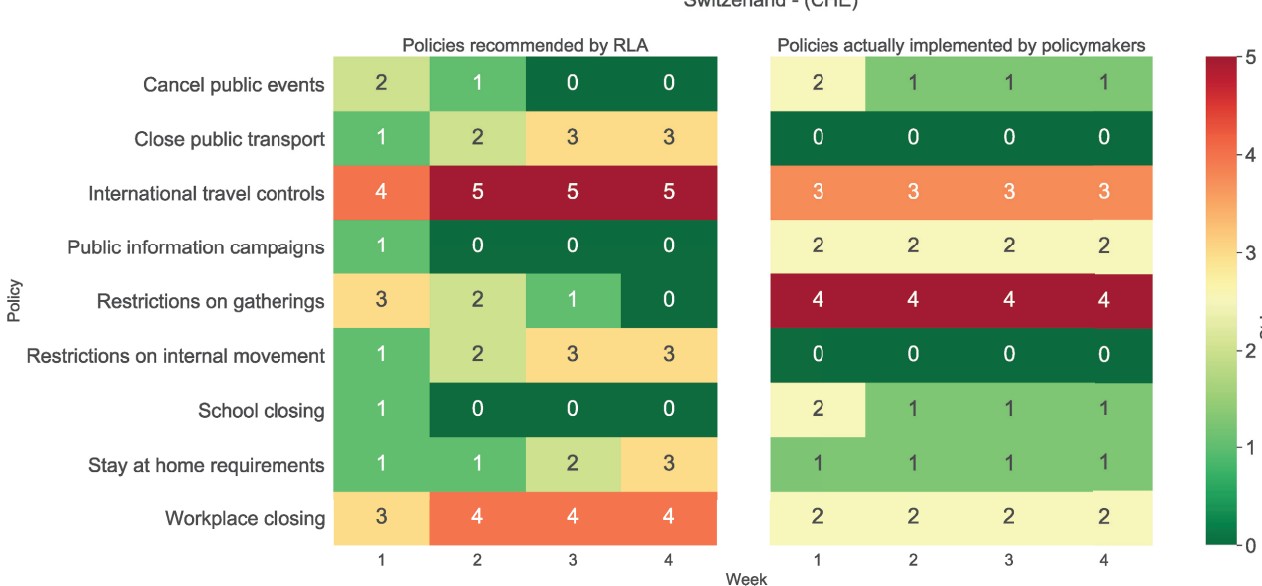

**Fig 8. Predicted best policy mix for Switzerland.** Starting from 12/04/2021 (included) for the following 28 days, compared to the actual implemented policies. Each of the nine policies are either not implemented (stringency of zero in green) or implemented in one of five levels of stringency (shaded green→yellow→red).

impact. It aims to provide insight into the costs and benefits of public health policies and create an analytical framework for guiding a country-specific optimised policy response. While it was trained on COVID-19 data, the framework could be adapted to future outbreaks with other pathogens or the public health response to non-infectious diseases.

The vast and diverse dataset used in this study allowed the models to attain high-accuracy, country-specific predictions on a large geographic scope. For instance, the estimates for "best policy combination" by our reinforcement learning agent achieved the same or better results (reduction in both $R_E$ and *UER*) than what was implemented in reality for every country investigated. Our results are unique in the literature, as the first public health policy decision support system suggesting the best customized policy mix tailored to the data of a specific country.

There are several limitations that must be mentioned. Firstly, it is important to note that this work specifically does not comment on the association between the metric predicted and the actual ground truth of that metric. That question would likely require large-scale coordinated primary data collection. Rather, we estimate how this reported metric (irrespective of its association with reality) would change in light of reported policy changes. As many policymakers use these metrics to measure relative policy effectiveness (usually with an understanding of their relationship to reality), our work thus attempts to align with the information that policymakers use routinely. Additionally, while our $R_E$ model covers approximately 90% of the global population, we restrict the *UER* model to countries where economic reporting is available and likely to be interoperable. More data is needed to extend the geographic scope of this model and we hope that the encouraging results from this work may help advocate for more regular and reliable reporting in lower resource settings.

As for all machine learning models, we could not predict random events. Indeed, the $R_E$ can be sensitive to super-spreader or localised transmission events, although it could be argued that these missed peaks do not represent transmission in the general population and are

therefore not under the direct control of public policies that are being investigated by the model. Another limitation is that our models assume legislative homogeneity within a country. However, this is certainly not the case for countries with autonomous sub-jurisdictions (i.e. decentralised state/provincial legislative bodies) where policies and transmission may vary considerably. While our approach can be adapted to finer-grained sub-geographies given appropriate data, we chose to report at a country level to maximise the number of included regions.

Lastly, errors in the predictions of $R_E$ and *UER* are reflected in the Reinforcement Learning Agent training. This could over- or under-estimate the efficacy of the chosen policies in absolute terms. However, we can argue that those policies still represent the best option, even if the reduction achieved in reality, in terms of $R_E$ and *UER*, when they are applied, is smaller than the predicted one.

Taken together, it is clear that policy decision support systems like this one are strictly not designed to replace expert opinion but rather to assist experts better understand the data on which they base their decisions.

## Conclusion

*What if. . .?* represents a new approach to guide policymakers in their decisions, whilst also supporting their strategy with objective data that facilitates communicating the logic of the intervention to the general population. The data-stream provides real time updates and we plan to release a public online platform in the near future. The platform will allow users to simulate hypothetical *What If. . .?* scenarios, and provide them with an evaluation of the policy response as well as a suggested "best" set of policies to adopt given their country's specific characteristics. The code is available in a fully open source repository at this link.

## Supporting information

**S1 Table. Data sources.**
(XLSX)

**S2 Table. Features of the $R_E$ and the unemployment[6] models.**
(XLSX)

**S3 Table. MSE of 116 countries when predicting their $R_E$.**
(XLSX)

**S4 Table. MSE of 29 countries when predicting their unemployment rate.**
(XLSX)

**S5 Table. MSE of 29 countries when predicting their $R_E$, reinforcement learning model.**
(XLSX)

**S6 Table. MSE of 29 countries when predicting their unemployment rate, reinforcement learning model.**
(XLSX)

**S7 Table. Reinforcement learning simulation results for the time period 07/12/2020–03/01/2021.** The initial value is the measured $R_E$ (left) *UER* (right) at the start of the period. The difference is the model performances calculated as measured value at the end of the period (i.e. as a result of policymaker) minus the one predicted by implementing the model's

recommended policies (higher is better, where positive values show the policy intervention of the RLA model improved the $R_E$ or *UER*).
(XLSX)

**S8 Table. Reinforcement learning simulation results for the time period 01/02/2021–28/02/2021.** The initial value is the measured $R_E$ (left) *UER* (right) at the start of the period. The difference is the model performances calculated as measured value at the end of the period (i.e. as a result of policymaker) minus the one predicted by implementing the model's recommended policies (higher is better, where positive values show the policy intervention of the RLA model improved the $R_E$ or *UER*).
(XLSX)

**S9 Table. Reinforcement learning simulation results for the time period 12/04/2021–09/05/2021.** The initial value is the measured $R_E$ (left) *UER* (right) at the start of the period. The difference is the model performances calculated as measured value at the end of the period (i.e. as a result of policymaker) minus the one predicted by implementing the model's recommended policies (higher is better, where positive values show the policy intervention of the RLA model improved the $R_E$ or *UER*).
(XLSX)

**S10 Table. Average difference between the measured final $R_E$/*UER* and the predicted one adopting the suggested policies.** Average obtained across three different time periods (07/12/2020–03/01/2021; 01/02/2021–28/02/2021; 12/04/2021–09/05/2021). Standard deviation is also reported.
(XLSX)

## Author Contributions

**Conceptualization:** Mary-Anne Hartley.

**Data curation:** Giorgio Mannarini, Francesco Posa, Thierry Bossy, Javier Fernandez-Castanon.

**Formal analysis:** Giorgio Mannarini, Francesco Posa, Thierry Bossy.

**Methodology:** Giorgio Mannarini, Francesco Posa.

**Project administration:** Martin Jaggi, Mary-Anne Hartley.

**Resources:** Javier Fernandez-Castanon.

**Software:** Giorgio Mannarini, Francesco Posa, Thierry Bossy, Lucas Massemin.

**Supervision:** Prakhar Gupta, Martin Jaggi, Mary-Anne Hartley.

**Validation:** Giorgio Mannarini, Francesco Posa, Tatjana Chavdarova, Pablo Cañas.

**Visualization:** Giorgio Mannarini, Francesco Posa, Thierry Bossy, Javier Fernandez-Castanon.

**Writing – original draft:** Thierry Bossy, Lucas Massemin, Tatjana Chavdarova, Pablo Cañas, Prakhar Gupta, Mary-Anne Hartley.

**Writing – review & editing:** Giorgio Mannarini, Francesco Posa, Prakhar Gupta, Martin Jaggi, Mary-Anne Hartley.

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
