## [Decision Letter · Decision Letter 0]

12 May 2022

PGPH-D-21-01132

What If...? Pandemic Policy-Decision-Support to Guide a Cost-Benefit-Optimised, Country-Specific Response

Dear Dr. Hartley,

Thank you for submitting your manuscript to PLOS Global Public Health. After careful consideration, we feel that it has merit but does not fully meet PLOS Global Public Health’s publication criteria as it currently stands. Therefore, we invite you to submit a revised version of the manuscript that addresses the points raised during the review process.

Please submit your revised manuscript by . If you will need more time than this to complete your revisions, please reply to this message or contact the journal office at globalpubhealth@plos.org. Please include the following items when submitting your revised manuscript:

We look forward to receiving your revised manuscript.

Kind regards,

Brooke E. Nichols

Academic Editor

Journal Requirements:

1. Please amend your Financial Disclosure statement. If you did not receive any funding for this study, please simply state: “The authors received no specific funding for this work.”

2. Please update your Competing Interests statement. If you have no competing interests to declare, please state: “The authors have declared that no competing interests exist.”

3. We notice that your supplementary tables are included in the manuscript file. Please remove them and upload them with the file type 'Supporting Information'. Please ensure that each Supporting Information file has a legend listed in the manuscript after the references list.

Additional Editor Comments (if provided):

Reviewers' comments:

Reviewer's Responses to Questions

**Comments to the Author**

1. Does this manuscript meet PLOS Global Public Health’s publication criteria? Is the manuscript technically sound, and do the data support the conclusions? The manuscript must describe methodologically and ethically rigorous research with conclusions that are appropriately drawn based on the data presented.

Reviewer #1: Yes

Reviewer #2: Yes

2. Has the statistical analysis been performed appropriately and rigorously?

Reviewer #1: I don't know

Reviewer #2: No

3. Have the authors made all data underlying the findings in their manuscript fully available (please refer to the Data Availability Statement at the start of the manuscript PDF file)?

Reviewer #1: Yes

Reviewer #2: Yes

4. Is the manuscript presented in an intelligible fashion and written in standard English?

Reviewer #1: Yes

Reviewer #2: Yes

5. Review Comments to the Author

Reviewer #1: The described application looks like one of the potentially more useful results of the pandemic, especially in that it has potential to be useful in future epidemics. I do not know enough about AI to be able to comment much on the validity of the methods.

My only comment is with regards to the data sources- as with any multi-country COVID-19 estimation project, the crux lies with being able to adjust for the systemic bias created by differences in the completeness of data reporting, in particular between LMIC and HIC. It is not clear to me how the authors corrected for this, in particular with regards to the case data informing R(e). (The efforts the authors took to exclude countries or periods with very low case data does not correct for this, as the problem does not automatically lead to very low cases reported, just to lower-than-should-be cases.) Similarly the accuracy of the OxCGRT data hinges on these restrictions having been publicised and their publication found by the OU researchers, which additionally introduces bias. One solution might be to declare this limitation and caution that the application will be differentially useful for HIV and LMIC, something we have in fact grown used to (writing from an LMIC).

Reviewer #2: Thank you to the authors for the manuscript and work involved in developing and analysing the modelling framework.

I have several major concerns with the model developed in the manuscript.

1. To develop an optimised cost-benefit response model for individual countries based on available data, is by definition not optimised. For many countries in the world, the data do not simply exist, or if they do are not necessarily representative, robust proxies of the situation they are seeking to measure. The authors realised this when developing the economic model where data were available for only 29 countries, and that too HICs. Decision-making in health is not only a function of health variables, environmental variables and 1 or 2 economic variables. Multi-tiered processes are in place with several stakeholders, international pressure often plays a large role, but perhaps the most important feature is that a country could have best 'optimised' policies, but implementation is out of the hands on the policymakers. Therefore, it is not possible to optimise a policy response without taking into account health infrastructure, implementation challenges, data system structure, public-private infrastructure and expert opinions where data are not available. It is for this reason that the premise of a central model to optimise a country's response fails.

2. Validating in retrospect

The finding of the analysis is the the model outperforms policymakers. I ask the authors, how valid such an analysis is in the first place. With the benefit of hindsight of months of data, to outperform actions that were based on little to no local data, lack of preparedness of systems that were gradually developed over time and global pressure for action, seems like a simple task, rather than evidence of superior intelligence of the model. If as the authors say weather is difficult to predict accurately, what chance does the model have when global pressure and scientific trends and politics will play a large role to inform the next set of policies?

3. Incomplete Data sources

COVID epidemiology is being characterised only by Re; a function of cases. This is dangerous, as the growth rates and Re do not necessarily take into account testing strategy, local test conditions and test sensitivity and most importantly hospital admissions, reported deaths and excess deaths. Training the model on Unemployment data for 29 countries (and that too HICs), and then stating the model to be globally relevant is a very big oversight.

4. Future application to COVID

By ignoring vaccination and new treatment options, the impact that changes in testing strategy has on Re, and higher seroprevalence brought on by previous infection and vaccination ( hybrid immunity), and most importantly by ignoring severe illness and death, this model is even less useful for future COVID public health modelling. Hybrid immunity means that one can expect less severe COVID (barring new variants with extreme immune loss), and with reduced testing, cases become a less relevant measure on which to base policy changes.

The model is not trained on sufficient public health data, mostly because it doesn't exist. The model does not incorporate contextual knowledge to allow it to be relevant for countries, particularly LMIC. While it may be of theoretical value, its usefulness in public health decision making is unfortunately limited. The manuscript may be better suited for publication in a theoretical statistical journal.

6. PLOS authors have the option to publish the peer review history of their article (what does this mean?). If published, this will include your full peer review and any attached files.

**Do you want your identity to be public for this peer review?** For information about this choice, including consent withdrawal, please see our Privacy Policy.

Reviewer #1: No

Reviewer #2: No

---

## [Editor Report · Decision Letter 1]

14 Jun 2022

What If...? Pandemic Policy-Decision-Support to Guide a Cost-Benefit-Optimised, Country-Specific Response

PGPH-D-21-01132R1

Dear Dr Hartley,

We are pleased to inform you that your manuscript 'What If...? Pandemic Policy-Decision-Support to Guide a Cost-Benefit-Optimised, Country-Specific Response' has been provisionally accepted for publication in PLOS Global Public Health.

Best regards,

Brooke E. Nichols

Academic Editor